# Treatment of COVID-19 during the Acute Phase in Hospitalized Patients Decreases Post-Acute Sequelae of COVID-19

**DOI:** 10.3390/jcm12124158

**Published:** 2023-06-20

**Authors:** Diana Badenes Bonet, Oswaldo Antonio Caguana Vélez, Xavier Duran Jordà, Merce Comas Serrano, Margarita Posso Rivera, Mireia Admetlló, Anna Herranz Blasco, Elisa Cuadrado Godia, Ester Marco Navarro, Gemma Martin Ezquerra, Zenaida Pineiro Aguin, Maria Cinta Cumpli Gargallo, Jose Gregorio Gonzalez Garcia, Eva Balcells Vilarnau, Diego Rodriguez Chiaradia, Xavier Castells, Joaquim Gea, Juan P. Horcajada, Judit Villar-García

**Affiliations:** 1Respiratory Medicine Department, Hospital del Mar, 08003 Barcelona, Spain; dbadenes@psmar.cat (D.B.B.); ocaguana@psmar.cat (O.A.C.V.); madmetllo@psmar.cat (M.A.); aherranzblasco@psmar.cat (A.H.B.); mcumpli@psmar.cat (M.C.C.G.); jggonzalezgarcia@psmar.cat (J.G.G.G.); ebalcells@psmar.cat (E.B.V.); darodriguez@psmar.cat (D.R.C.); jgea@psmar.cat (J.G.); 2Department of Medicine and Life Sciences, Universitat Pompeu Fabra (UPF), 08003 Barcelona, Spain; zpineiro@psmar.cat (Z.P.A.); jhorcajada@psmar.cat (J.P.H.); 3Hospital del Mar Medical Research Institute (IMIM), 08003 Barcelona, Spain; xduran@imim.es; 4Assessoria Metodològica i Bioestadística (AMIB), 08003 Barcelona, Spain; 5Epidemiology and Evaluation Department, Hospital del Mar, REDISSEC, RICAPPS, 08003 Barcelona, Spain; mcomas@psmar.cat (M.C.S.); mposso@psmar.cat (M.P.R.); xcastells@psmar.cat (X.C.); 6Neurology Department, Hospital del Mar, 08003 Barcelona, Spain; ecuadrado@psmar.cat; 7Physical Medicine and Rehabilitation Department, Hospital del Mar, 08003 Barcelona, Spain; emarco@psmar.cat; 8Dermatology Department, Hospital del Mar, 08003 Barcelona, Spain; gmartin@psmar.cat; 9Otorrinolaringology Department, Hospital del Mar, 08003 Barcelona, Spain; 10Centro de Investigación en Red de Enfermedades Respiratorias, (CIBERES), Instituto de Salud Carlos III (ISCIII), 08003 Barcelona, Spain; 11Infectious Diseases Department, Hospital del Mar, 08003 Barcelona, Spain

**Keywords:** post-acute sequelae of COVID-19 (PASC), long COVID, SARS-CoV-2, remdesivir, corticosteroids

## Abstract

Background: The post-acute sequelae of SARS-CoV-2 (PASC) infection have caused a significant impact on our health system, but there is limited evidence of approved drugs focused on its prevention. Our objective was to identify risk factors that can determine the presence of PASC, with special attention to the treatment received in the acute phase, and to describe the profile of persistent symptoms in a multidisciplinary Post-Coronavirus Disease-19 (COVID-19) Unit. Methods: This one-year prospective observational study included patients following an acute COVID-19 infection, irrespective of whether they required hospital admission. A standardized symptom questionnaire and blood sampling were performed at the first follow-up visit, and demographic and clinical electronic data were collected. We compared subjects with PASC with those who had fully recovered. Multivariate logistic regression was performed to identify factors associated with PASC in hospitalized patients, and Kaplan–Meier curves were used to assess duration of symptoms according to disease severity and treatments received in the acute phase. Results: 1966 patients were evaluated; 1081 had mild disease, 542 moderate and 343 severe; around one third of the subjects had PASC, and were more frequently female, with obesity, asthma, and eosinophilia during acute COVID-19 disease. Patients who received treatment with dexamethasone and remdesivir during the course of the acute illness showed a lower median duration of symptoms, compared with those who received none of these treatments. Conclusion: Treatment with dexamethasone and/or remdesivir may be useful to reduce the impact of PASC secondary to SARS-CoV-2 infection. In addition, we identified female gender, obesity, asthma, and disease severity as risk factors for having PASC.

## 1. Introduction

Post-Acute sequelae of SARS-CoV-2 infection (PASC) currently affects millions of people worldwide [1,2]. Even in mild cases of SARS-CoV-2 infection, medium and long term health problems have been described, leading to significant health resource expenditure [3]. The lack of a consistent definition for PASC or long-term COVID-19 makes comparison between studies difficult. To address these challenges, a universal definition has been proposed, emphasizing symptoms persisting beyond one month (mild cases) or three months (severe cases) with a significant impact on quality of life [4]. Studies on long COVID-19 have revealed various findings related to immunology, virology, vascular issues, neurological symptoms, and myalgic encephalomyelitis/chronic fatigue syndrome. Multi-organ damage has been observed, with a significant overlap in symptoms [5]. Consequently, different studies have been published describing persistent symptoms in approximately one third of the subjects, with dyspnea, fatigue, exercise intolerance, joint pain, headache, ‘brain fog’ and psychological symptoms being the main reported ones [3,6,7]. Therefore, there is a growing interest in identifying the risk factors of developing PASC. Several publications have identified disease severity, female sex, age, body mass index, the presence of asthma and the number of symptoms during the first week of disease as potential factors that could be used to estimate the individual’s risk for PASC [8,9]. Moreover, a study identified an immunoglobulin (Ig) signature, based on total IgM and IgG3 levels, which, combined with other factors such as age, history of asthma and symptoms during the primary infection, was able to predict the risk of PASC independently of the time-point of blood sampling [10].

Despite the important impact of PASC on our health systems, there is a lack of studies specifically designed to evaluate whether treatments administered in the acute phase of COVID-19 disease can reduce the risk of PASC. There is evidence from a study that demonstrated that patients treated with interferon β-1b and antiviral therapy in the acute phase showed a greater probability of recovering their initial health status [11]. Furthermore, there is growing evidence supporting the effectiveness of antiviral therapies in reducing the risk of PASC. Specifically, the use of molnupiravir within five days of a positive SARS-CoV-2 test result has been associated with a reduced risk of PASC, post-acute death, and post-acute hospital admission in individuals with at least one risk factor for severe COVID-19 [12]. Similarly, the combined antivirals nirmatrelvir–ritonavir have also shown a reduction in the risk of PASC, post-acute death, and post-acute hospital admission in outpatients [13].

For now, there are no specific drug recommendations centered on PASC prevention. Current approved treatments, such as interleukin-6 receptor antagonists and dexamethasone, have been demonstrated to improve outcomes during hospitalization and survival during the acute infection in moderate and severe cases [14,15]. Moreover, new antiviral therapies are recommended for patients with risk of progression to severe COVID-19 since they can also reduce hospitalizations and deaths [16]. However, PASC can also occur in young people without risk factors.

On this point, and to understand the PASC prevalence, its risk factors and the potential impact of treatment received better, it is imperative to appropriately design follow-up strategies optimizing health resources.

Thus, our primary objective was to identify potential risk factors that can determine the presence and duration of PASC, with a special focus on the treatment received during the acute phase. Our second objective was to describe persistent symptoms and their duration in a multidisciplinary Post-COVID-19 Unit.

## 2. Materials and Methods

### 2.1. Study Design

This is a single-center prospective observational study carried out according to the principles of the Declaration of Helsinki for human investigations and approved by the local Ethics Committee (nº 2020/9455). Patients with a positive SARS-CoV-2 PCR test were offered the chance of participating at the first follow-up visit, and were then finally included after giving their verbal informed consent. A standardized symptom questionnaire (Appendix A) was administered in the first follow-up visit to all the participants and demographic and clinical electronic data were collected. Blood samples were obtained from outpatients and from subjects who required hospitalization. Finally, participants were evaluated every 3 months and symptom persistence was evaluated and registered (Appendix A).

### 2.2. Study Participants

#### 2.2.1. Inclusion Criteria

We included patients who had suffered from mild disease (not hospitalized) and those individuals with mild, moderate, or severe pneumonia due to COVID-19 (all of them admitted to our hospital) and were visited in our multidisciplinary Post-COVID-19 unit from June to December 2020 and followed-up during 1 year. Those who required hospitalization were followed in the Post-COVID-19 Unit, irrespective of whether they had persistent symptoms, whereas patients referred from primary care were symptomatic. Inclusion criteria were to have had a COVID-19 infection, confirmed with polymerase chain reaction (PCR), during the first and second wave of the pandemic in our center (March to December 2020). The predominant SARS-CoV-2 variants that circulated at that time were lineage B.1.1.7 (88%) followed by B. 1.177/A222V (7.5%). This study was conducted during the early phases of the COVID-19 vaccination programs and most of our patients had not been vaccinated at the time of data collection.

#### 2.2.2. Exclusion Criteria

Subjects who refused to participate, who died during the follow-up, as well as those who we were unable to contact after hospital discharge were excluded.

### 2.3. COVID-19 Severity Groups

Participants were stratified by the acute COVID-19 infection severity into (a) mild disease: patients without need of supplemental oxygen therapy, (b) moderate disease: hospitalized patients who required supplemental low-flow oxygen therapy (FiO_2_ < 0.40) and (c) severe disease: hospitalized patients with supplemental high-flow oxygen therapy (FiO_2_ > 0.40), high-flow nasal cannula oxygen therapy (HFNC) or non-invasive and/or invasive mechanical ventilation [17].

### 2.4. Treatment Received for SARS-CoV-2 Infection

In our center, treatment modality has been protocolized since April 2020. Antivirals were indicated (remdesivir 200 mg the first day and 100 mg/day for 5 days) to patients with low-flow oxygen requirements to maintain SpO_2_ > 94%. In addition, treatment with heparin was indicated at a prophylactic dose of 40 mg/day in patients with D-dimer levels < 2000 mcg/L, intermediate doses of 1 mg/Kg/day with D-dimer levels > 2000 mcg/L, and anticoagulant doses of 1 mg/Kg/12 h in patients diagnosed with acute pulmonary embolism. Subjects with moderate–severe disease received treatment with dexamethasone 6 mg/day for 10 days and tocilizumab was posteriorly administered only in cases of worsening of respiratory symptoms, FiO_2_ > 35% and IL-6 levels > 40 pg/mL, according to our hospital protocol and available guidelines at that moment [18,19]. Moreover, subjects with organizing pneumonia received treatment with methylprednisolone that was progressively tapered at follow-up. Outpatients included in the study received no therapy.

### 2.5. Symptom Assessment

A structured questionnaire (including 22 symptoms) was performed at the initial visit (Appendix A) by trained interviewers of the Post-COVID-19 Unit, and symptom persistence was registered at each follow-up visit to record symptom duration. The presence of PASC was considered according to the current definitions at the beginning of the pandemic, including ≥2 persistent symptoms affecting any organ or system after 4 weeks of the acute infection, having excluded other potential etiologies [20,21].

### 2.6. Blood Sampling

In hospitalized subjects, the main data from blood analyses, both at baseline and at the peak of disease severity, were collected. Moreover, another blood test was performed at inclusion at the first follow-up visit. In outpatients, only one blood sample was collected at the inclusion. The blood test parameters included C-reactive protein, ferritin, procalcitonin, interleukin-6 (IL-6), leucocyte count, creatinine, lactate dehydrogenase (LDH), glucose, total proteins, D-dimer, and platelets.

### 2.7. Statistical Analysis

Categorical variables were expressed as numbers and percentages, whereas continuous variables were presented as mean (standard deviation, SD), or median (25th and 75th percentiles, P25–P75) when the normality assumption was not fulfilled. Differences between groups were checked with Pearson’s Chi Squared test for categorical variables, and unpaired Student’s *t*-test or one-way ANOVA for continuous variables meeting the normality assumption, or non-parametric Mann–Whitney U test when normality could not be assumed. Significant differences were considered at the 0.05 *p* level. Multivariate logistic regression was performed to check factors associated with PASC in hospitalized patients. Variables in the multivariate analysis were introduced based on clinical criteria considering those that were statistically significant or nearly significant (*p*, 0.5–0.1) in the bivariate analysis. Results from this analysis were reported as odds ratios (OR) with their 95% confidence interval (CI). Kaplan–Meier curves were used to compare duration of symptoms by disease severity and the effects of treatments used in the acute phase. The log-rank test was performed to assess differences in symptom duration for each drug between its use or not in every group of disease severity. STATA version 15.1 (StataCorp, College Station, TX, USA) was used for statistical analysis.

## 3. Results

The study cohort included 2716 subjects referred after hospitalization and from primary care following a confirmed SARS-CoV-2 infection. Subjects who died or were lost during the follow-up (416 and 334, respectively) were excluded. In total, 1966 patients were finally followed-up at the Post-COVID-19 Unit (Figure 1).

### 3.1. Baseline Characteristics

The clinical characteristics and persisting symptoms of the total population are summarized in Table 1. The mean age was 56.4 years, and 50% were females. As far as disease severity is concerned, 54.9% had mild disease, whereas 27.6 and 17.5% showed moderate or severe disease, respectively. The most frequent comorbidity was arterial hypertension, followed by psychiatric disorders and diabetes mellitus, and all comorbidities were more frequent in subjects with severe disease. Around a third of the subjects experienced persisting symptoms at the follow-up, this being more frequent in the severe disease group. Dyspnea, fatigue and myalgia were the most symptomatic manifestations.

### 3.2. PASC in Hospitalized Patients

#### 3.2.1. Clinical Data

Comparing subjects with PASC with those that completely recovered after hospitalization (Table 2), the former were older, more frequently females and had a higher frequency of obesity and asthma than the latter. COVID-19 severity was also associated with a higher PASC frequency. The most frequent persisting symptoms were again dyspnea, fatigue and myalgia, followed by cough, anxiety and depression.

#### 3.2.2. Laboratory Data

Subjects with PASC had a significantly lower total lymphocyte count, lower Cd4 and Cd8, and higher interleukin 6 (IL-6), lactate dehydrogenase (LDH) and platelet count at baseline than those without. Laboratory results at the peak of the acute disease evidenced that the PASC group showed higher neutrophil, lymphocyte, eosinophil and platelet counts, as well as LDH levels, than subjects that completely recovered (Table 3).

#### 3.2.3. PASC Prediction

Table 4 summarizes the results of the multivariate logistic regression model to identify predictors of PASC in hospitalized patients. In our cohort, female sex, severe disease, and asthma increased the risk of having PASC. Since the most severe patients usually show greater changes in inflammatory parameters, our multivariate analysis included this factor to avoid a potential bias. Although results were not statistically significant, a strong trend for having a higher risk of PASC was observed for eosinophil and neutrophil counts at the peak point of the acute disease.

### 3.3. PASC in Outpatients

A total of 233 patients did not require hospitalization but were followed-up at the Post-COVID-19 Unit of our center (Table 5). Those subjects with PASC were more frequently female and had more psychiatric disorders than those without, and the symptoms more frequently displayed were dyspnea, fatigue and myalgia, again followed by cough, anxiety and depression. No statistically significant differences were found in the follow-up laboratory parameters (Appendix A).

### 3.4. Treatment Received

Given that treatment received is highly dependent on disease severity, we analyzed the effect of treatment with this in mind. In those cases that received treatment with remdesivir with a moderate or severe disease, a significant shorter duration of PASC was observed when compared with those not receiving the treatment (109 vs. 262 and 187 vs. 284 days, respectively). Similar results were found in patients who received dexamethasone, in cases with moderate or severe disease (133 vs. 271 and 266 vs. 309, respectively Table 6, Figure 2). Treatments with tocilizumab and methylprednisolone were not associated with a reduction in the duration of PASC symptoms.

Some patients received a combination of some of these therapies, according to disease severity. In total, 603 (71.2%) patients only received corticosteroids (dexamethasone, prednisone or methylprednisolone); 79 (9.3%) patients received corticosteroids and remdesivir and 187 (22.1%) patients received corticosteroids and tocilizumab (Table 6).

## 4. Discussion

This prospective study systematically described the profile of PASC one year after an acute COVID-19 infection in a large group of patients with a broad scope of clinical severity. Even more relevant is that, to our knowledge, this is the first prospective study in patients who required oxygen therapy and hospital admission to show that some treatments used during the acute phase reduces the risk of PASC. This is an important point, since the current approved treatments of COVID-19 infection are focused on preventing acute complications in people with risk factors, and preventing PASC in mild cases in an outpatient setting. In our study, we make a distinct contribution by focusing on a study cohort consisting of patients who presented in the acute phase of COVID-19, encompassing a range of severity from mild to moderate and severe cases. Our dataset includes data from hospitalized patients, providing valuable insights into the long-term health outcomes of this specific population. This comprehensive approach allows us to capture a broader spectrum of patients and provide a more comprehensive understanding of the long-term effects of COVID-19.

Indeed, a very large number of people at risk of PASC are relatively young without any comorbidities, so they currently do not meet the criteria for receiving treatment during the acute phase. However, there is very limited evidence on the association between treatments administered during the acute phase and predisposition of persistent symptoms [22,23,24,25]. There is an Italian cohort, where patients were treated in the acute phase of COVID-19 with monoclonal drugs, which obtained a 56% reduction in the risk of PASC appearance [26]; unlike our cohort, they required no oxygen therapy or hospital admission for SARS-CoV-2 infection. Our study shows that treatment with remdesivir and/or corticosteroids during the acute phase decreases the length of PASC in hospitalized subjects. Furthermore, corticosteroids or antivirals are more accessible worldwide than monoclonal drugs.

Remdesivir has been shown to be effective in slowing disease progression and shortening recovery time for COVID-19 patients [27], with the antiviral treatment being recommended for the patients with COVID-19 infection in national and international clinical guidelines, including WHO, NIH, NICE, ESCMID, IDSA, and SEIMC [18,19,28,29,30,31], although it is recommended to be restricted to patients with up to 7 days of symptoms, mild disease, but with no risk factors for progression to severe disease, and also in patients with moderate or severe infection requiring hospitalization [18,20]. The results of different studies led certain guidelines to recommend prolonging its administration in severe cases requiring invasive ventilation or ECMO or in patients with severe immunosuppression, since persistent viral replication beyond two weeks has been described in these groups [32]. The persistence of viral replication has also been involved in the pathophysiology of PASC. Therefore, it does not seem strange that the early administration of antiviral treatment can also prevent long-term symptoms [27,33]. Furthermore, some of the drugs currently in use to treat acute COVID-19 infection have potential anti-inflammatory and immunomodulatory effects that have already been analyzed in several cohorts and meta-analyses [22,34,35]. Those treatments, mainly tocilizumab and corticosteroids, are used for patients in the cytokine storm phase, and, although they have managed to reduce mortality when appropriately indicated [36], it is unknown to date whether they could also reduce the presence or length of PASC. However, immunoactivation and/or sustained inflammation have also been involved in the pathophysiology of PASC. Thus, it also seems reasonable that these treatments could be effective in reducing the impact of PASC. In fact, the inflammatory cascade that remains activated in certain susceptible individuals when they overcome the acute phase of COVID-19 infection includes cytokines and signaling molecules (such as interleukins 1β, -6, -8, -17 and tumor necrosis factor alpha) that have been involved in structural and functional abnormalities in respiratory and limb muscles, potentially partly explaining some PASC symptoms such as musculoskeletal discomfort, fatigue and dyspnea, which are among the most common persisting symptoms [37,38].

In the present study, we found that treatment with dexamethasone in the acute phase reduced the duration of PASC. However, treatment with methylprednisolone was not associated with a reduction in the long-term duration of symptoms, probably because it was intended to treat patients with severe pneumonia who were in the organizing/fibrosis phase. In the case of treatment with tocilizumab, we were unable to demonstrate effects on PASC. One possible hypothesis is that this would be explained by the action of this drug on a very specific pathway of IL6 and the use of a single dose with a short half-life.

According to the results of this study, and given both the high prevalence of PASC, which in many cases is disabling, and the burden that its care entails for the health system, administering treatment during the acute phase should be considered for all patients.

In this prospective study, we found that close to a quarter of the patients had at least one persistent symptom of COVID-19 months after the acute infection, and this is similar to the prevalence found in other series [6,9,16,39]. The same accounts for the main symptoms, which were dyspnea, fatigue, myalgia, persistent cough, anxiety and depression, were also similar to those described in several previous studies [16,40,41,42].

When comparing the present cohort with other series, the risk factors for PASC are relatively similar. In general, these studies identified age, female sex, comorbidities and severity of the disease during the acute process [43,44]. Age is probably an important predisposing factor, probably due to the lower regenerative capacity of the lung, in addition to underlying chronic pulmonary problems that could be associated [44,45]. A recent meta-analysis of patients identified several epidemiological and clinical risk factors associated with an increased risk of developing persistent symptoms of COVID-19, known as Post-COVID Conditions (PCC). Female sex, older age, higher BMI, smoking and preexisting comorbidities such as anxiety, depression, asthma, COPD, diabetes, IHD and immunosuppression were found to be significant risk factors for PCC. Hospitalization or ICU care during the acute phase of COVID-19 infection also increased the risk. These findings highlight the multifactorial nature of PCC and provide insights into its risk factors [46]. On the other hand, vaccination was found to have a protective role against PCC [46,47,48,49]. In our study, the severity of the acute disease was a predisposing factor for presenting PASC, related to the requirement for invasive and non-invasive respiratory support, as with other series [50,51,52,53,54]. Interestingly, asthma and obesity were the comorbidities that were associated with a higher risk of persistent symptoms after acute infection, regardless of the severity of the disease in hospitalized patients. These findings have also been found in other studies [8,10]. More severe COVID-19 disease has consistently been associated with both asthma and obesity. While the predisposition of the former has been related to increased concentrations of IL-4 [55,56]. Other frequent comorbidities also described in other series [6,9] were arterial hypertension, diabetes mellitus, COPD and anxiety-depressive disorders, but they did not show significant results in the present study, except in outpatients, in which subjects with PASC were more frequently female and had more psychiatric disorders than those without PASC.

Some laboratory findings linked to an inflammatory profile have also been found to increase the risk for PASC, such as a significant decrease in total lymphocyte counts—especially CD4+ T cells and CD8+ T cells—and an increase in LDH, CRP and IL-6 levels, as well as IL-1β, IL-6, TNF and IP10 [5,51,57,58,59,60], mostly in patients who required hospitalization. This could be due to the inflammatory and immunological pathophysiological mechanisms involved in the COVID-19 disease [61,62]. Moreover, our results also highlight the potential role of eosinophil counting during the acute disease in hospitalized patients. Counts of these cells were higher in our patients with PASC, and showed a tendency to be an independent factor of presenting persisting symptoms, even in the multivariate analysis, where variables that may be confounding factors were included. Some series have shown that, during the early phases of the acute phase, the levels of eosinophils are rather low, but could rise at 3 and 6 months [63,64,65]. Eosinophils play an important role in immunity; they eliminate bacteria and viruses, as well as behaving as an antigen-presenting cell [66]. However, the presence of eosinophils is also related to autoimmune and allergic diseases, which is consistent with the pathophysiology of COVID-19 and PASC [67] and may also justify the increased risk in patients with asthma. One of the hypotheses for the persistence of symptoms is the hyperactivation of the autoimmunity that the acute infection generates [68]. To our knowledge, this is also the first study that confers a prognostic value on the presence of eosinophils during acute infection, since it is related to persistent symptoms. No statistically significant differences were found in the follow-up laboratory parameters in outpatients, but in this group of patients the analytical study was not performed during the primary infection.

## 5. Strengths and Limitations

Our study investigates the impact of treatment during the acute phase on the risk of PASC and presents several strengths and limitations. One of the major strengths is that it is one of the few reports to highlight the potential benefit of treatment in the acute phase in reducing the risk of PASC. What is new is that the treatment decreases the incidence of PASC, including in patients without risk factors for severe COVID-19, for whom treatment is not currently recommended. This emphasizes the importance of treating patients who may not develop severe COVID-19 but are at risk for PASC. Additionally, our study benefits from a large and diverse population sample, encompassing individuals with varying severity levels of COVID-19, and a comprehensive and lengthy follow-up period that includes both clinical and laboratory parameters. Moreover, the standardized treatment protocols implemented since April 2020 have facilitated a more homogeneous comparison, with reliable data collected from electronic health records.

Our study has certain limitations that need to be acknowledged. Firstly, it is possible that patients who did not participate had milder symptoms compared to those who did, which could lead to an overestimation of the prevalence of prolonged symptoms, particularly among non-hospitalized individuals with mild COVID-19. Additionally, being a single-center study conducted during the early phase of the pandemic, the generalizability of our findings to patients infected with later variants of SARS-CoV-2 may be uncertain. We acknowledge that the lack of consideration for the specific viral variants circulating during the study period is a limitation of our research. In addition, the study was carried out during the initial period of the pandemic, before widespread vaccination campaigns. Therefore, the analysis did not include information on the vaccination status of the patients, which could have influenced treatment outcomes and tissue response. Moreover, it is important to recognize the absence of a control group as a notable limitation. However, it is important to note that despite this limitation, the study still provides valuable insights into the potential effects of the treatments on PASC. The findings should be interpreted cautiously, considering the absence of a control group [69].

## 6. Conclusions

Treatment with dexamethasone and remdesivir was associated with a shorter duration of symptoms after SARS-CoV-2 infection. In a 12-month follow-up more than one third of the subjects had persistent symptoms. We identified female sex, obesity, asthma, severity of the disease and eosinophilia as risk factors associated with the presence of PASC. In summary, early identification of people at a higher risk of PASC and an early initiation of antiviral and/or anti-inflammatory treatment may avoid the risk of long-term complications and reduce the burden that PASC places on the healthcare system. Further multicentric studies are needed to identify risk factors of PASC as well as to establish which therapies could reduce its risk.

## Figures and Tables

**Figure 1 jcm-12-04158-f001:**
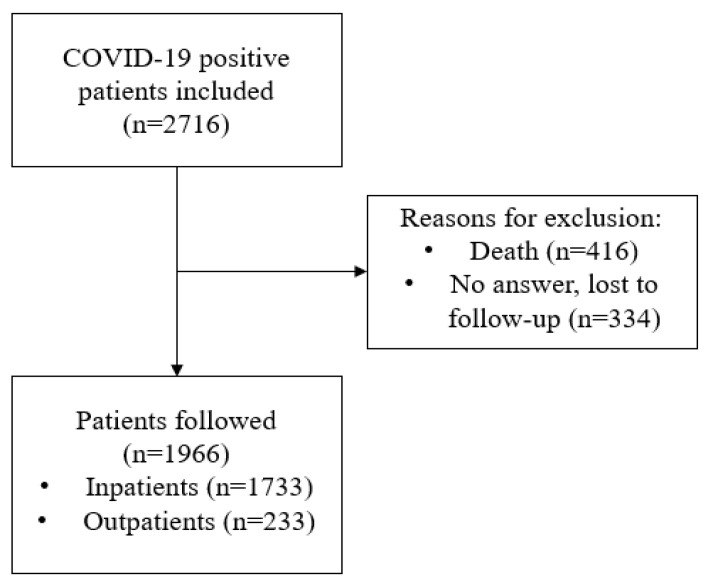
Flow diagram of subjects evaluated in the Post-COVID-19 Unit.

**Figure 2 jcm-12-04158-f002:**
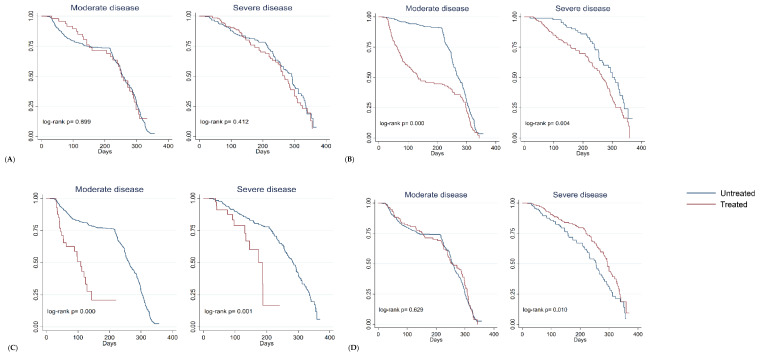
Kaplan-Meier survival curves that integrate the different treatments administered or not during the acute phase, which compares the median duration days of persistent symptoms, according to the severity of acute COVID-19. Patients with moderate disease *n* = 542 (27.6%), with severe disease *n* = 343 (17.5%), all patients *n* = 1966. (**A**) Tocilizumab: the median duration of symptoms was 252 days vs. 226 with moderate disease and 269 days vs. 292 with severe disease; (**B**) Dexamethasone: the median duration of symptoms was 133 days vs. 271 with moderate disease and 266 days vs. 309 with severe disease. (**C**) Remdesivir: the median duration of symptoms was 109 days vs. 262 with moderate disease and 187 days vs. 284 with severe disease; (**D**) Methylprednisolone: the median duration of symptoms was 252 days vs. 226 with moderate disease and 294 days vs. 255 with severe disease.

**Table 1 jcm-12-04158-t001:** Clinical characteristics and persisting symptoms of the total population according to acute COVID-19 severity.

	All Patients *n* = 1966	Mild *n* = 1081 (54.9%) (No Supplemental Oxygen FiO_2_ 21%)	Moderate *n* = 542 (27.6%) (Supplemental Oxygen FiO_2_ < 40%)	Severe *n* = 343 (17.5%) (FiO_2_ > 40%, HFNC, NIV, IMV)	*p* Value
**Demographics**					
Age (years), mean (SD)	56.4 (17.1)	51 (16.5)	63 (15.8)	64 (14.0)	**<0.001**
Females, *n* (%)	974 (49.5)	616 (56.9)	228 (42.1)	130 (37.9)	**<0.001**
**Comorbidities, *n* (%)**					
Arterial hypertension	578 (29.4)	181 (16.7)	235 (43.3)	162 (47.2)	**<0.001**
Psychiatric disorders	379 (19.2)	182 (16.8)	117 (21.6)	80 (23.3)	0.348
Diabetes Mellitus	258 (13.1)	81 (7.5)	108 (19.9)	69 (20.1)	**<0.001**
Obesity	131 (6.6)	36 (3.3)	49 (9.0)	46 (13.4)	**<0.001**
Kidney failure	166 (8.4)	43 (3.9)	60 (11.1)	63 (18.3)	**<0.001**
COPD	122 (6.2)	36 (3.3)	52 (9.6)	34 (9.9)	**<0.001**
Asthma	120 (6.1)	64 (5.9)	38 (7.0)	18 (5.2)	0.492
Heart failure	61 (3.1)	11 (1.0)	28 (5.2)	22 (6.4)	**<0.001**
Neoplasia	46 (2.3)	14 (1.3)	20 (3.7)	12 (3.5)	**0.003**
Solid organ transplant	11 (1.0)	1 (0.9)	4 (0.7)	6 (1.7)	**0.004**
**Persisting symptoms, *n* (%)**	728 (37.0)	370 (34.2)	164 (30.2)	194 (56.6)	**<0.001**
Dyspnea	472 (24.0)	230 (21.3)	108 (19.9)	134 (39.1)	**<0.001**
Fatigue	391 (19.8)	208 (19.2)	88 (16.1)	95 (27.7)	**<0.001**
Myalgia	237 (12.0)	126 (11.6)	47 (8.7)	64 (18.6)	**<0.001**
Anxiety symptoms	227 (11.5)	139 (12.8)	48 (8.8)	40 (11.6)	**0.035**
Depressive symptoms	198 (10.0)	115 (10.6)	44 (8.1)	39 (11.3)	0.081
Memory loss	177 (8.9)	103 (9.5)	39 (7.2)	35 (10.2)	0.087
Motor dysfunction	162 (8.2)	72 (6.6)	32 (5.9)	58 (17.0)	**0.001**
Palpitations	129 (6.5)	84 (7.7)	21 (3.9)	24 (7.0)	**0.008**
Hair loss	124 (6.3)	64 (5.9)	22 (4.1)	38 (11.1)	**<0.001**
Anosmia/hyposmia	121 (6.2)	79 (7.3)	25 (4.6)	17 (5.0)	**0.040**
Headache	119 (6.0)	86 (8.0)	16 (3.0)	14 (4.1)	**<0.001**
Persistent cough	117 (5.9)	67 (6.2)	20 (3.7)	30 (8.7)	**0.004**
Ageusia/hypogeusia	106 (5.4)	65 (6.0)	21 (3.9)	20 (5.8)	0.083
Hypoesthesia	103 (5.2)	55 (5.1)	15 (2.8)	33 (9.6)	**<0.001**
Dizziness/instability	91 (4.6)	46 (4.2)	20 (3.7)	25 (7.3)	**0.015**
Audition loss	73 (3.7)	42 (3.8)	15 (2.8)	16 (4.6)	0.120
Reduced mobility	61 (3.1)	26 (2.4)	17 (3.1)	18 (5.2)	**0.015**
Chest pain	56 (2.8)	29 (2.7)	14 (2.6)	13 (3.8)	0.168
Dysphagia	52 (2.6)	24 (2.2)	14 (2.6)	14 (4.1)	0.080
Skin alterations/mouth ulcers	68 (3.5)	39 (3.6)	13 (2.4)	16 (4.6)	0.076
Orthopnea	38 (1.9)	19 (1.8)	10 (1.8)	9 (2.6)	0.189
Fever	30 (1.5)	24 (2.2)	3 (0.6)	3 (0.9)	**0.013**

Abbreviations: FiO_2_, oxygen inspiratory fraction; HFNC, high-flow nasal cannula; NIV, non-invasive ventilation; IMV, invasive mechanical ventilation; COPD, chronic obstructive pulmonary disease. Statistical significance (*p* < 0.05) is indicated by bold font.

**Table 2 jcm-12-04158-t002:** Characteristics of COVID-19 hospitalized patients with and without PASC (*n* = 1733).

	Without PASC (*n* = 1389)	With PASC (*n* = 344)	*p* Value
**Demographics**			
Age (years), mean (SD)	56.5 (17.5)	58.6 (15.6)	0.047
Females, *n* (%)	624 (44.9)	208 (60.5)	**<0.001**
**Comorbidities, *n* (%)**			
Arterial hypertension	440 (31.7)	126 (36.6)	0.135
Psychiatric disorders	293 (21.1)	83 (24.1)	0.278
Diabetes Mellitus	202 (14.5)	52 (15.1)	0.835
Obesity	91 (6.6)	39 (11.3)	**0.004**
Kidney failure	130 (9.4)	33 (9.6)	0.978
COPD	91 (6.6)	30 (8.7)	0.179
Asthma	78 (5.6)	38 (11.0)	**<0.001**
Heart failure	46 (3.3)	14 (4.1)	0.593
Neoplasia	38 (2.7)	7 (2.0)	0.459
Solid organ transplant	6 (0.4)	5 (1.5)	0.058
**Disease severity, *n* (%)**			**<0.001**
Mild	754 (53.7)	143 (41.0)	
Moderate	426 (30.3)	97 (27.8)	
Severe	224 (16.0)	109 (31.2)	
**Persisting symptoms, *n* (%)**			
Dyspnea	164 (12.4)	187 (56.7)	**<0.001**
Fatigue	108 (8.2)	175 (53.0)	**<0.001**
Myalgia	56 (4.2)	107 (32.4)	**<0.001**
Persistent cough	41 (3.1)	43 (13.0)	**<0.001**
Anxiety symptoms	70 (5.3)	89 (27.0)	**<0.001**
Depressive symptoms	61 (4.6)	74 (24.0)	**<0.001**
Memory loss	65 (4.9)	56 (17.0)	**<0.001**
Hair loss	57 (4.3)	28 (8.5)	**<0.002**
Headache	34 (2.6)	44 (13.3)	**<0.001**
Hypoesthesia	32 (2.4)	36 (10.9)	**<0.001**
Ageusia/hypogeusia	39 (2.9)	31 (9.4)	**<0.001**
Anosmia/hyposmia	46 (3.5)	38 (11.5)	**<0.001**
Motor dysfunction	63 (4.8)	59 (17.6)	**<0.001**
Dysphagia	21 (1.6)	17 (5.1)	**<0.001**
Chest pain	24 (1.8)	14 (4.2)	**0.009**
Audition loss	24 (1.8)	26 (7.8)	**<0.001**
Dizziness/instability	33 (2.5)	34 (10.1)	**<0.001**
Reduced mobility	26 (1.9)	26 (7.8)	**<0.001**
Orthopnea	20 (1.5)	11 (3.3)	**0.003**
Palpitations	34 (2.5)	53 (15.8)	**<0.001**
Skin alterations/mouth ulcers	24 (1.8)	13 (3.9)	**0.020**
Fever	7 (0.5)	8 (2.4)	**<0.001**

Abbreviations: PASC, post-acute sequelae of COVID-19; COPD, chronic obstructive pulmonary disease. Statistical significance (*p* < 0.05) is indicated by bold font.

**Table 3 jcm-12-04158-t003:** Laboratory data of patients evaluated in hospital setting (*n* = 1445).

Blood Test Data, Mean (SD) *	Without PASC (*n* = 1159)	With PASC (*n* = 286)	*p* Value
**Baseline value ****			
Glucose (mg/dL)	124.77 (53.32)	127.68 (54.98)	0.414
Creatinine (mg/dL)	0.97 (0.53)	0.99 (0.87)	0.526
LDH (U/L)	282.79 (110.88)	299.66 (125.97)	**0.029**
Ferritin (ng/mL), median (p25–p75)	696.50 (367.00–1180.50)	764.53 (261.00–1203.50)	0.743
IL-6 (pg/mL), median (p25–p75)	23.63 (7.60–57.62)	26.50 (11.40–75.42)	**0.008**
Procalcitonin (ng/mL)	0.70 (5.14)	0.33 (1.54)	0.400
CRP (mg/dL)	7.79 (7.82)	7.77 (7.63)	0.968
Total proteins (g/dL)	6.48 (0.61)	6.42 (0.59)	0.178
Total Lymphocytes (×10^3^/μL)	1.23 (0.84)	1.08 (0.55)	**0.020**
Lymphocytes T (×10^3^/μL)	1.28 (0.85)	1.20 (0.61)	0.130
CD4-T Lymphocyte (×10^3^/μL)	525.22 (320.77)	477.34 (292.01)	0.065
CD8-T Lymphocyte (×10^3^/μL)	286.37 (213.68)	253.05 (179.30)	0.051
Coefficient CD4/CD8	2.34 (1.60)	2.39 (1.92)	0.719
Neutrophils (×10^3^/μL)	5.32 (3.22)	5.37 (2.93)	0.811
Eosinophils (×10^3^/μL)	0.05 (0.09)	0.04 (0.08)	0.449
D-dimer (μg/L), median (p25–p75)	630 (390–1050)	600 (400–1040)	0.612
Platelets (×10^3^/μL)	221.62 (85.71)	236.62 (93.01)	**0.010**
**Peak value ****			
Glucose (mg/dL)	116.20 (58.32)	116.90 (57.92)	0.861
Creatinine (mg/dL)	1.04 (0.70)	1.11 (1.02)	0.169
LDH (U/L)	304.75 (123.48)	332.70 (144.98)	**0.001**
Ferritin (ng/mL), median (p25–p75)	723 (393–972)	693 (308–951)	0.688
IL-6 (pg/mL), median (p25–p75)	16.60 (4.80–53.40)	25.40 (6.50–71)	0.120
Procalcitonin (ng/mL)	0.72 (4.41)	0.67 (1.83)	0.905
CRP (mg/dL)	6.31 (5.72)	6.28 (5.43)	0.924
Total proteins (g/dL)	6.63 (0.57)	6.59 (0.51)	0.334
Total Lymphocytes (×10^3^/μL)	1.31 (1.04)	1.16 (0.60)	**0.050**
Lymphocytes T (×10^3^/μL)	1.82 (1.02)	1.97 (0.91)	**0.019**
CD4-T Lymphocyte (×10^3^/μL)	554.12 (340.29)	513.64 (329.85)	0.137
CD8-T Lymphocyte (×10^3^/μL)	300.81 (222.68)	272.28 (187.62)	0.101
Coefficient CD4/CD8	2.41 (1.62)	2.48 (1.98)	0.625
Neutrophils (×10^3^/μL)	611.07 (1658.34)	990.23 (2753.85)	**0.029**
Eosinophils (×10^3^/μL)	0.13 (0.19)	0.19 (0.28)	**<0.001**
D-dimer (μg/L), median (p25–p75)	710 (440–950)	740 (460–915)	0.814
Platelets (×10^3^/μL)	325.19 (141.74)	365.57 (150.91)	**<0.001**

* Data expressed as mean and standard deviation unless otherwise specified. Abbreviations: PASC, post-acute sequelae of COVID-19; LDH, Lactate dehydrogenase; IL-6, Interleukin 6; CRP, c-reactive protein. ** Baseline laboratory values correspond to those obtained at hospital admission. Peak laboratory values correspond to the maximum clinical severity. Statistical significance (*p* < 0.05) is indicated by bold font.

**Table 4 jcm-12-04158-t004:** Multivariate logistic regression model of PASC at follow-up in subjects that required hospitalization.

*n* = 1432	OR (95% CI)	*p*-Value
**Age**	1.00 (0.99, 1.01)	0.862
**Gender**		
Male	1.00	
Female	2.01 (1.51, 2.67)	**<0.001**
**Asthma**		
No	1.00	
Yes	2.04 (1.30, 3.20)	**0.002**
**Obesity**		
No	1.00	
Yes	1.35 (0.88, 2.08)	0.170
**Psychiatric disorders**		
No	1.00	
Yes	1.11 (0.81, 1.51)	0.528
**COVID-19 severity**		
Mild	1.00	
Moderate	1.38 (0.97, 1.96)	0.074
Severe	2.48 (1.63, 3.79)	**<0.001**
**Urea (peak), (mg/dL)**		
Urea (peak) < 49	1.00	
Urea (peak) ≥49	1.02 (0.71, 1.46)	0.915
**Lymphocytes T (peak), (** **×10^3^/μL)**		
Lymphocytes T (peak) < 3	1.00	
Lymphocytes T (peak) ≥ 3	1.09 (0.69, 1.74)	0.715
**Eosinophils T (peak), (** **×10^3^/μL)**		
Eosinophils T (peak) < 0.5	1.00	
Eosinophils T (peak) ≥ 0.5	1.75 (0.99, 3.12)	0.056
**Neutrophils (peak), (** **×10^3^/μL)**		
Neutrophils (peak) < 7	1.00	
Neutrophils (peak) ≥ 7	1.37 (0.98, 1.92)	0.068
**Platelets (peak), (** **×10^3^/μL)**		
Platelets (peak) < 410	1.00	
Platelets (peak) ≥ 410	1.15 (0.84, 1.57)	0.382

Abbreviations: PASC, post-acute COVID-19 syndrome, OR: Odds Ratio; 95% CI: 95% confidence interval. Peak laboratory values correspond to the situation of maximum clinical severity. Statistical significance (*p* < 0.05) is indicated by bold font.

**Table 5 jcm-12-04158-t005:** Characteristics of COVID-19 outpatients with and without PASC (*n* = 233).

	Without PASC (*n* = 112)	With PASC (*n* = 121)	*p* Value
**Demographics**			
Age (years), mean (SD)	52.30 (15.86)	52.26 (15.05)	0.981
Females, *n* (%) Males, *n* (%)	52 (46.4%) 60 (53.5%)	90 (74.4%) 31 (25.6%)	<0.001 <0.001
**Comorbidities, *n* (%)**			
Arterial hypertension	7 (6.3%)	5 (4.1%)	0.465
Psychiatric disorders	19 (18.1%)	44 (38.9%)	**0.001**
Diabetes Mellitus	2 (1.8%)	2 (1.7%)	0.938
Obesity	0 (0.0%)	1 (12.5%)	0.191
Kidney failure	2 (15.4%)	1 (12.5%)	0.854
COPD	1 (0.9%)	0 (0.0%)	0.298
Asthma	2 (15.4%)	2 (25.0%)	0.586
Heart failure	1 (7.7%)	0 (0.0%)	0.421
Neoplasia	1 (0.9%)	0 (0.0%)	0.298
Solid organ transplant	0 (0%)	0 (0%)	
**Persisting symptoms, *n* (%)**			
Dyspnea	43 (41.0%)	78 (69.0%)	**<0.001**
Fatigue	27 (25.7%)	81 (71.7%)	**<0.001**
Myalgia	18 (17.1%)	56 (49.6%)	**<0.001**
Persistent cough	10 (9.5%)	23 (20.4%)	**0.026**
Anxiety symptoms	14 (13.3%)	54 (47.8%)	**<0.001**
Depressive symptoms	19 (18.1%)	44 (38.9%)	**0.001**
Memory loss	14 (13.3%)	42 (37.2%)	**<0.001**
Hair loss	20 (19.0%)	19 (16.8%)	0.667
Headache	10 (9.5%)	31 (27.4%)	**0.001**
Hypoesthesia	11 (10.5%)	24 (21.2%)	**0.031**
Ageusia/hypogeusia	18 (17.1%)	18 (15.9%)	0.809
Anosmia/hyposmia	19 (18.1%)	18 (15.9%)	0.670
Motor dysfunction	19 (18.1%)	21 (18.6%)	0.926
Dysphagia	5 (4.8%)	9 (8.0%)	0.335
Chest pain	6 (5.7%)	12 (10.6%)	0.189
Audition loss	12 (11.4%)	12 (10.6%)	0.849
Dizziness/instability	6 (5.7%)	19 (16.8%)	**0.010**
Reduced mobility	1 (1.0%)	9 (8.0%)	**0.013**
Orthopnea	1 (1.0%)	6 (5.3%)	0.068
Palpitations	11 (10.5%)	34 (30.1%)	**<0.001**
Skin alterations/mouth ulcers	12 (11.4%)	19 (16.8%)	0.255
Fever	1 (1.0%)	15 (13.3%)	**<0.001**

Abbreviations: PASC, post-acute COVID-19 syndrome; COPD, chronic obstructive pulmonary disease. Statistical significance (*p* < 0.05) is indicated by bold font.

**Table 6 jcm-12-04158-t006:** Median duration of symptoms in days, according to COVID-19 severity and treatments administered during the acute phase.

		Moderate Disease		Severe Disease	
Treatments		*n*	p_50_ (p_25_–p_75_) *	*n*	p_50_ (p_25_–p_75_)
Remdesivir	No	472	262 (217, 304)	296	284 (216, 336)
	Yes	43	109 (45, 143)	36	187 (131, 188)
Tocilizumab or Sarilumab	No	466	256 (143, 303)	193	292 (216, 336)
	Yes	49	252 (150, 298)	139	269 (175, 325)
Methylprednisolone or Prednisone	No	395	256 (138, 300)	126	255 (158, 312)
	Yes	120	252 (153, 308)	206	294 (220, 336)
Dexamethasone	No	285	271 (236, 308)	87	309 (239, 341)
	Yes	230	133 (63, 298)	245	266 (158, 330)

* Data expressed as median of days and interquartile range [p_50_ (p_25_–p_75_)] unless otherwise specified.

## Data Availability

The data presented in this study are available on request from the corresponding author.

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
