# Peer review of "Treatment of COVID-19 during the Acute Phase in Hospitalized Patients Decreases Post-Acute Sequelae of COVID-19"

_jcm, 2023, doi:10.3390/jcm12124158_

Round 1

Reviewer 1 Report

We live a difficult moment in the history of humanity in the period between 2020-2022.

Our understanding of this pandemic will come with time and the scientific work that will be produced.

This manuscript was well prepared and very important, since it associates the treatment performed on the patient, according to the severity of the condition and the patient's evolution in post Covid- 19

Best regards

Author Response

Assigned Editor: Thea Wu

Journal of Clinical Medicine

Manuscript Status: Pending major revisions

Manuscript ID:  jcm-2387544

TITLE:

Treatment of Covid-19 during the Acute Phase in Hospitalized Patients Decreases Post-Acute Sequelae of Covid-19.

Response to Reviewer 1 Comments

Point 1: We live a difficult moment in the history of humanity in the period between 2020-2022.

Our understanding of this pandemic will come with time and the scientific work that will be produced.

This manuscript was well prepared and very important, since it associates the treatment performed on the patient, according to the severity of the condition and the patient's evolution in post Covid- 19

Best regards

Response 1:  Thank you for your insightful comments and suggestions regarding our manuscript. We appreciate your recognition of the importance of this study in the context of the challenging period humanity faced between 2020 and 2022.

We agree with your observation that tailoring treatments based on the severity of the illness is crucial. Our findings support this notion, emphasizing the need to individualize treatment strategies according to the specific needs of patients. This approach not only optimizes patient outcomes but also contributes to our understanding of effective treatment strategies for COVID-19. Once again, we appreciate your thoughtful comments and suggestions, and we will carefully consider them in revising our manuscript.

Sincerely,

  1. Antonio Caguana Velez

Reviewer 2 Report

This manuscript supports the idea that patients who received treatment with dexamethasone and remdesivir during  
acute illness showed a lower median duration of symptoms compared with those who received 36 none of these treatments. There is a tremendous bias in the design of this study. It is well documented that not necessarily the prognostic of Covid-19 patients was exclusively associated with the antiviral treatment. Dexamethasone alone in other trials resulted in similar results.

Author Response

Assigned Editor: Thea Wu

Journal of Clinical Medicine

Manuscript Status: Pending major revisions.

Manuscript ID:  jcm-2387544

TITLE:

Treatment of Covid-19 during the Acute Phase in Hospitalized Patients Decreases Post-Acute Sequelae of Covid-19.

Response to Reviewer 2 Comments

Point 1: This manuscript supports the idea that patients who received treatment with dexamethasone and remdesivir during acute illness showed a lower median duration of symptoms compared with those who received 36 none of these treatments. There is a tremendous bias in the design of this study. It is well documented that not necessarily the prognostic of Covid-19 patients was exclusively associated with the antiviral treatment. Dexamethasone alone in other trials resulted in similar results.

Response 1:  Thank you for your valuable comments on our manuscript. Considering the concerns regarding potential bias in the study design, .it is important to consider that our study was conducted during a global pandemic, and we had to follow international protocols and recommendations at that time. The study design was influenced by the limitations and constraints faced during the pandemic, including the availability of medications. Moreover, the objectives of our study included also describing the profile of persistent symptoms in a multidisciplinary Post-COVID-19 Unit as well as to identify different risk factors that could determine the presence of PASC. 

We will include a discussion of these limitations in our manuscript to provide a balanced interpretation of the results and promote further investigation into the impact of different treatments on COVID-19 outcomes.

While acknowledging the limitations, our study provides valuable insights reflecting the real-world experience during that period.

We acknowledge that the prognosis of patients with COVID-19 is influenced by various factors beyond antiviral treatment. We did not claim that antiviral treatment was the sole determinant of prognosis in COVID-19. In fact, we identified also in our manuscript other risk factors associated to PASC like disease severity, obesity and asthma which are common to other publications on the field. We appreciate the reviewer's comment, as it emphasizes the importance of considering other factors that may influence COVID-19 prognosis.

On the other hand regarding the suggestion about the potential role of dexamethasone in reducing the risk of PASC; to our knowleg dexamethasone has been shown to improve survival in selected cases and could influence reducing hospital stay (https://doi.org/10.1016/j.farma.2023.02.001), but there is no direct evidence that reduces the risk of PASC and therefore, more research is needed to investigate the specific impact of dexamethasone on PASC.

However, our findings highlight the importance of optimizing treatment strategies for the effective prevention and management of post-COVID-19 symptoms.

We sincerely apologize not including the strengths and limitations of our study in the initial manuscript explaining the complexity in which it was conducted and the likely biases.

We greatly appreciate your feedback, as it has prompted us to address this important aspect. We have now taken your suggestion into consideration and have added a section on strengths and limitations to the revised version of our manuscript.

(See modification line 392-418, page 14). 5. Strengths and limitations. Our study investigates the impact of treatment during the acute phase on the risk of PASC and presents several strengths and limitations. One of the major strengths is being one of the few reports to highlight the potential benefit of treatment in the acute phase in reducing the risk of PASC. What is new is that the treatment decreases the PASC, includ-ing in patients without risk factors for severe COVID-19, for whom treatment is not cur-rently recommended. This emphasizes the importance of treating patients who may not develop severe COVID-19 but are at risk for PASC. Additionally, our study benefits from a large and diverse population sample, encompassing individuals with varying severity levels of COVID-19, and a comprehensive and lengthy follow-up period that includes both clinical and laboratory parameters. Moreover, the standardized treatment protocols im-plemented since April 2020 have facilitated a more homogeneous comparison, with relia-ble data collected from electronic health records.

Our study has certain limitations that need to be acknowledged. Firstly, it is possible that patients who did not participate had milder symptoms compared to those who did, which could lead to an overestimation of the prevalence of prolonged symptoms, particu-larly among non-hospitalized individuals with mild COVID-19. Additionally, being a single-center study conducted during the early phase of the pandemic, the generalizability of our findings to patients infected with later variants of SARS-CoV-2 may be uncertain. We acknowledge that the lack of consideration for the specific viral variants circulating during the study period is a limitation of our research. In addition, the study was carried out during the initial period of the pandemic, before widespread vaccination campaigns. Therefore, the analysis did not include information on the vaccination status of the pa-tients, which could have influenced treatment outcomes and tissue response. Moreover, it is important to recognize the absence of a control group as a notable limitation. The lack of a control group hinders direct comparison between treated and untreated individuals, making it challenging to establish a causal relationship between the administered treat-ments and the observed effects on PASC.

Once again, we sincerely thank you for your valuable feedback.

Sincerely,

O.Antonio Caguana Vélez

Reviewer 3 Report

Major concerns;

1-     The novelty  and the significance of the study are low.

2-     The study design (June2020-Dec2021) vaccination status of patients in late period was not studied which is important for the tissue response for treatment.

Other concerns;

- Introduction: You keep writing (recently, recent publications,..) however the references were published in 2021 may not be considered recent in regards to COVID-19 era.

“our primary objective was to identify potential risk factors that can determine 73 the presence and duration of PASC, with special focus on the treatment received in the 74 acute phase. Our second objective was to describe persistent symptoms and their duration 75 in a multidisciplinary Post-COVID-19 Unit.” Why then did you state in the title only (TREATMENT)?

Spelling mistakes are seen (eg line60)

-Methods: Is it for one year or one and half?

- Inclusion and exclusion criteria should be clearly mentioned in methods

-Why do not you classify age as age groups and test?

- In results; values and units in table3 should be revised.

- In table 5 why male patients not included?

- In discussion you should discuss all significant results.

- Refer to the limitations of the study

Some mistakes are detected, proofreading is highly recommended 

Author Response

Assigned Editor: Thea Wu

Journal of Clinical Medicine

Manuscript Status: Pending major revisions.

Manuscript ID:  jcm-2387544

TITLE: Treatment of Covid-19 during the Acute Phase in Hospitalized Patients Decreases Post-Acute Sequelae of Covid-19.

Response to Reviewer 3 Comments

Point 1: The novelty and the significance of the study are low.

Response 1:  We respectfully address the concerns raised regarding the novelty and significance of our study. We believe our study brings valuable contributions to the field of Post-Acute Sequelae of SARS-CoV-2 infection (PASC) research.

First, the size of our population sample is substantial, providing a robust foundation for our findings. This large sample allows for a more comprehensive understanding of the impact of treatment during the acute phase on PASC.

Furthermore, while it is true that there have been retrospective studies on PASC, our study stands out as a prospective investigation conducted in a dedicated multidisciplinary unit specifically designed for managing PASC. This specialized setting enables a more comprehensive and focused evaluation of PASC, ensuring reliable and accurate data collection.

Importantly, our study's novelty lies in demonstrating that treatment during the acute phase can reduce the risk of PASC, even in individuals without known risk factors for severe COVID-19. This finding challenges the current recommendations that primarily focus on treating only severe cases, highlighting the potential benefits of early intervention in preventing PASC among individuals with milder infections.

By showing that treatment can mitigate the development of PASC, our study has significant implications for clinical practice. It emphasizes the importance of considering treatment options for individuals who may not develop severe COVID-19 but are at risk of experiencing debilitating PASC symptoms.

We would like to inform the reviewer that we have taken their feedback into consideration and made appropriate changes to address the strengths of our study in the manuscript.

(See modification line 390-403, page 14): 5. Strengths and limitations. Our study investigates the impact of treatment during the acute phase on the risk of PASC and presents several strengths and limitations. One of the major strengths is being one of the few reports to highlight the potential benefit of treatment in the acute phase in reducing the risk of PASC. What is new is that the treatment decreases the PASC, includ-ing in patients without risk factors for severe COVID-19, for whom treatment is not cur-rently recommended. This emphasizes the importance of treating patients who may not develop severe COVID-19 but are at risk for PASC. Additionally, our study benefits from a large and diverse population sample, encompassing individuals with varying severity levels of COVID-19, and a comprehensive and lengthy follow-up period that includes both clinical and laboratory parameters. Moreover, the standardized treatment protocols im-plemented since April 2020 have facilitated a more homogeneous comparison, with relia-ble data collected from electronic health records….

Point 2: The study design (June2020-Dec2021) vaccination status of patients in late period was not studied which is important for the tissue response for treatment.

Response 2:  Thank you for your valuable comments on our manuscript regarding the vaccination status of the study participants.

We understand the importance of considering the vaccination status of patients in relation to treatment outcomes, particularly in the context of tissue response. However, it is important to note that our study was conducted between June 2020 and December 2021, during a period when vaccination programs against COVID-19 were still in their early stages, and most of our patients had not been vaccinated at the time of initiation. Also, at the beginning of the pandemic the indirect effects of the vaccines and the variants of the SARS COV-2 virus were unknown, therefore the objective of collecting data was not considered. As you correctly mentioned, currently the vaccination status of patients can influence the immune response and subsequent tissue reactions to treatment. We recognize that the impact of vaccination on treatment outcomes in PASC is an important area of research that warrants further investigation.

In light of his comment, we will include a clarification in the revised manuscript to clearly indicate that our study specifically refers to patients who were mostly unvaccinated at the time of data collection. This clarification will ensure transparency and prevent any misinterpretation of our findings in relation to the general vaccinated population.

(See modification line 115-117 page 3): This study was conducted during the early phases of the COVID-19 vaccination programs and most of our patients had not been vaccinated at the time of data collection.

Point 3: Introduction: You keep writing (recently, recent publications,..) however the references were published in 2021 may not be considered recent in regards to COVID-19 era.

Response 3: Thank you very much for the corrections, we have made the change and the correction.

(See modification line 54, page 2): Consequently, different studies have been published recently describing persistent symp-toms in approximately one third of the subjects, with dyspnea, fatigue, exercise intoler-ance, joint pain, headache, ‘brain fog’ and psychological symptoms being the main re-ported ones..

(See modification line 58, page 2): Recent Several publications have identified disease severity, female sex, age, body mass index, the presence of asthma and the number of symptoms during the first week of dis-ease as potential factors that could be used to estimate the individual’s risk for PASC..

(See modification line 61, page 2): Moreover, a recent study identified an immunoglobulin (Ig) signature, based on total IgM and IgG3 levels, which, combined with other factors such as age, history of asthma, and symptoms during the primary infection, was able to predict the risk of PASC, inde-pendently of the time-point of blood sampling.

(See modification line 287, page 12): There is only an Italian cohort, where patients were treated in the acute phase of COVID-19

We have added new bibliographic references recommended by the reviewers.

Point 4: “our primary objective was to identify potential risk factors that can determine 73 the presence and duration of PASC, with special focus on the treatment received in the 74 acute phase. Our second objective was to describe persistent symptoms and their duration 75 in a multidisciplinary Post-COVID-19 Unit.” Why then did you state in the title only (TREATMENT)?

Response 4:  We appreciate the reviewer's comment on the title of our manuscript. We acknowledge that our initial title focused primarily on the aspect of treatment during the acute phase as something new. However, we understand the importance of accurately reflecting the objectives of the study in the title. After considering the reviewer's suggestion, we have revised the title to better encompass the scope of our research and may be more appropriate to the objectives of the proposed study as follows: " Risk factors and Impact of Treatment during the Acute Phase in Hospitalized Patients in Post-Acute Sequelae of Covid-19: A one-year prospective study", another option is "Assessment of risk factors and the influence of acute phase treatment on the post-acute sequelae of Covid-19: a one-year prospective study".

We thank the reviewer for this valuable feedback, and we have made the necessary adjustments to better align the title with the study's objectives.

Point 5: Spelling mistakes are seen (eg line60)

Response 5:  Thank you for your comment regarding the spelling mistakes. We appreciate your feedback and apologize for any errors that may have been present in the manuscript. We have thoroughly reviewed the document to identify and correct any spelling mistakes or typographical errors. Your input has been valuable in ensuring the quality of our work, and we strive to deliver an error-free manuscript.

(See modification line 67-69, page 2): There is evidence from a study that demonstrated that patients treated with interferon β-1b and antiviral therapy in the acute phase showed a greater probability of recovering their initial health status.

(See modification line 84-86, page 2): On this point, and  to understand the PASC prevalence, its risk factors and the potential impact of treatment received better, it is imperative to appropriately design follow-up strategies optimizing health resources.

(See modification line 334-336, page 13): The same accounts for the main symptoms, which were dyspnea, fatigue, myalgia, persistent cough, anxiety, and depression, also like those described in several previous studies.

(See modification line 352, page 13): support, like to other series [52-56]. Interestingly, asthma and obesity were the

(See modification line 373, page 14): but could rise at to 3 and 6 months [65-67]. Eosinophils play an important role in.

Point 6: Methods: Is it for one year or one and half?

Response 6:  Thank you for pointing out the need for clarification regarding the duration in the Methods section. We apologize for any confusion caused. To clarify, the inclusion of patients in our study included a period of 7 months; from June 2020 to December 2020. However, the follow-up period for each patient was one year. Therefore, we conducted a one-year follow-up of the patients included in the study, starting from the time of their visit to our multidisciplinary Post-COVID-19 unit. We appreciate the reviewer's attention to detail and have made the necessary clarification in the manuscript to accurately reflect the timeline of patient inclusion and follow-up.

(See modification line 105-107, page 3): Patients who had suffered from mild disease (not hospitalized) and those individuals with mild, moderate, or severe pneumonia due to COVID-19 (all of them admitted to our hospital) and were visited in our multidisciplinary Post-COVID-19 unit from June to December 2020 and followed-up during 1 year.

Point 7: Inclusion and exclusion criteria should be clearly mentioned in methods

Response 7:  We acknowledge the importance of providing a thorough description of these criteria to ensure transparency and reproducibility of the study. We appreciate the reviewer's feedback, and we address this issue by revising the Methods section. By doing so, we aim to provide a comprehensive and detailed account of the patient selection process. Thank you for bringing this to our attention, and we will ensure that the revised manuscript includes this important information.

(See modification line 103-120, page 3): 2.2. Study participants

Inclusion criteria:

Patients who had suffered from mild disease (not hospitalized) and those individuals with mild, moderate, or severe pneumonia due to COVID-19 (all of them admitted to our hospital) and were visited in our multidisciplinary Post-COVID-19 unit from June to December 2020 and followed-up during 1 year.. Those who required hospitalization were followed in the Post-COVID-19 Unit, irrespective of whether they had persistent symptoms, whereas patients referred from primary care were symptomatic. Inclusion criteria were to have had a COVID-19 infection, confirmed with polymerase chain reaction (PCR), during the first and second wave of the pandemics in our center (March to December 2020). The predominant SARS-CoV-2 variants that circulated at that time were lineage B.1.1.7 (88%) followed by B. 1.177/A222V (7.5%). This study was conducted during the early phases of the COVID-19 vaccination programs and most of our patients had not been vaccinated at the time of data collection.

     Exclusion criteria:

Subjects who refused to participate, who died during the follow-up as well as those who we were unable to contact after hospital discharge were excluded.

Point 8: Why do not you classify age as age groups and test?

Response 8:  We appreciate the reviewer's comment regarding the classification of age as age groups and its association with PASC. In our study, we did classify age into different groups for analysis, and we performed logistic regression to explore the potential association between age groups and PASC.

Table 4 presents the results of the multivariate logistic regression model, including various factors as potential predictors of PASC in hospitalized patients. We observed that age did not show a statistically significant association with PASC in this model (OR = 1.00, 95% CI: 0.99-1.01, p = 0.862). While the p-value suggests no significant association, it is important to note that the odds ratio is very close to 1.00, indicating minimal effect of age on the likelihood of developing PASC. Otherwise, it has been widely explained in the literature the convenience in using continuous variables as continuous or categorical taking into account the distribution of the continuous variable or the loss of information when any variable is binned among other criteria.

(Ref*: Problems Caused by Categorizing Continuous Variables. https://discourse.datamethods.org/t/categorizing-continuous-variables/3402)

On the other hand, blood determination parameters were entered as dichotomized due to the skewed sampling distribution of these parameters. Anyway, and as sensitivity analysis, we have performed the same analysis displayed on table 4, but binning the age in quartiles (see adjunct table).

Considering these findings, it appears that age alone may not be a strong independent predictor of PASC among hospitalized patients in our study.

Table 4:  With age in terciles.

OR (95% CI)

p-value

Age

<=48

1.00

49 - 60

1.17 (0.78 - 1.75)

0.444

61 - 72

1.35 (0.89 - 2.05)

0.154

>72

1.07 (0.68 - 1.68)

0.757

Point 9: In results; values and units in table 3 should be revised.

Response 9:  Thank you for your valuable feedback.  Values and units presented in Table 3 of the manuscript have been modified. We appreciate you bringing this to our attention. We have carefully reviewed and revised the table to ensure accurate and consistent reporting of the values and units. The corrected version of Table 3 will be included in the revised manuscript.

(See modification line 211-215, page 7-8): table 3

Point 10: In table 5 why male patients not included?

Response 10:  Thank you for bringing up the concern regarding the inclusion of male patients in Table 5. We apologize for any confusion caused. Male patients were indeed included in the study, and their data are expressed in the table. The formatting or representation of the table fragment we provided may have created the impression that male patients were not included. Specifically, the proportion of males is significantly different between the two groups (without PASC and with PASC), with a lower percentage of males observed in the group with PASC compared to the group without PASC.

We have made the necessary adjustments to the table and ensure that it accurately represents the data in our manuscript.

(See modification line 235, page 9)

Table 5. Characteristics of COVID-19 outpatients with and without PASC (n = 233).

Without PASC                   (n = 112)

With PASC                  (n = 121)

p value

Demographics

 Age (years), mean (SD)

52.30 (15.86)

52.26 (15.05)

0.981

 Females, n (%)

                 Males, n (%)

52 (46.4%)

60 (53.57)

90 (74.4%)

31 (25.62)

<0.001

<0.001

Comorbidities, n (%)

Point 11:  In discussion you should discuss all significant results.

Response11:  Thank you for your feedback.  Based on your comment, we have carefully revised and modified the discussion to ensure that all significant findings are adequately addressed. Additionally, we have incorporated relevant studies that contribute to the understanding of our results. We thank you for your valuable input, and we are committed to incorporating any other observations or suggestions you may have to further enhance the quality and comprehensiveness of our manuscript.

(See modification line 268-383) page 13-14.

  1. Discussion

(See modification line 273-282) page 13: This is an important point since the current approved treatments of COVID-19 infection are focused on preventing acute complications in people with risk factors, and preventing PASC in mild cases in outpatient setting. In our study, we make a distinct contribution by focusing on a study cohort consisting of patients who presented in the acute phase of COVID-19, encompassing a range of severity from mild to moderate and severe cases. Our dataset includes data from hospitalized patients, providing valuable insights into the long-term health outcomes of this specific population. This comprehensive approach allows us to capture a broader spectrum of patients and provide a more comprehensive understanding of the long-term effects of COVID-19.

(See modification line 339-349) page 13: Age is probably an important predisposing factor, being probably due to the lower regen-erative capacity of the lung, in addition to underlying chronic pulmonary problems that could be associated [4639,4740]. A recent meta-analysis of patients identified several epi-demiological and clinical risk factors associated with an increased risk of developing per-sistent symptoms of COVID-19, known as Post-COVID Conditions (PCC). Female sex, older age, higher BMI, smoking, and preexisting comorbidities such as anxiety, depres-sion, asthma, COPD, diabetes, IHD, and immunosuppression were found to be significant risk factors for PCC. Hospitalization or ICU care during the acute phase of COVID-19 in-fection also increased the risk. These findings highlight the multifactorial nature of PCC and provide insights into its risk factors [48]. On the other hand, vaccination was found to have a protective role against PCC.

(See modification line 357-361) page 13: Other frequent comorbidities also described in other series [6,9] were arterial hyperten-sion, diabetes mellitus, COPD, and anxiety-depressive disorders, but they have not shown significant results in the present study, except in outpatients, in which subjects with PASC were more frequently female and had more psychiatric disorders than those without PASC.

(See modification line 357-361) page 13: Some laboratory findings linked to an inflammatory profile have also been found to increase the risk for PASC such as the significant decrease in total lymphocyte counts es-pecially CD4+ T cell, CD8+ T cell and an increase in LDH, CRP and IL-6 levels as well as IL-1β, IL-6, TNF and IP10.

Point 12: Refer to the limitations of the study

Response 12:  We apologize for the omission of the limitations of our study in the initial version of the manuscript. We have made the necessary revisions to the manuscript and included a section on the limitations of our study. We thank you for bringing this to our attention, and we are grateful for your valuable input in improving the quality of our research.

(See modification line  404-418, page 14): Our study has certain limitations that need to be acknowledged. Firstly, it is possible that patients who did not participate had milder symptoms compared to those who did, which could lead to an overestimation of the prevalence of prolonged symptoms, particularly among non-hospitalized individuals with mild COVID-19. Additionally, being a single-center study conducted during the early phase of the pandemic, the generalizability of our findings to patients infected with later variants of SARS-CoV-2 may be uncertain. We acknowledge that the lack of consideration for the specific viral variants circulating during the study period is a limitation of our research. In addition, the study was carried out during the initial period of the pandemic, before widespread vaccination campaigns. Therefore, the analysis did not include information on the vaccination status of the patients, which could have influenced treatment outcomes and tissue response. Moreover, it is important to recognize the absence of a control group as a notable limitation. The lack of a control group hinders direct comparison between treated and untreated individuals, making it challenging to establish a causal relationship between the administered treatments and the observed effects on PASC.

We acknowledge that the submission process for this article has been lengthy on our part, resulting in the inclusion of references from earlier publications. We also recognize that significant scientific advancements regarding symptoms and the condition of prolonged COVID-19 have emerged in recent months. Despite these challenges, we appreciate the valuable feedback provided by the reviewers and have made efforts to address their comments and incorporate the most relevant and up-to-date references available at the time of revision.

Once again, we sincerely thank you for your valuable feedback.

Sincerely,

O.Antonio Caguana Vélez

Reviewer 4 Report

The paper presents some flaws.

It lacks important references, for instance: https://pubmed.ncbi.nlm.nih.gov/36639608/; https://pubmed.ncbi.nlm.nih.gov/36951832/

It is without a control group, an important limitation of many studies on Long Covid. See: https://pubmed.ncbi.nlm.nih.gov/36951832/

The attrition rate is not specified (lines 100/101).

The definition of Long Covid in the study is debatable, not in accordance with many guidelines (NICE, WHO, CDC; see : https://pubmed.ncbi.nlm.nih.gov/36708796/).

Morevoer, a definition as the one proposed here would be better: https://pubmed.ncbi.nlm.nih.gov/37017975/

The role of variant is not addressed.

Strange figures are reported in the tables: e.g., in table 5, subjects without PASC onset have dyspnea in 41% at follow-up. Quite high rate for a "non-PASC" group.

Lines 263-265 completely erroneous, since there are many studies on the effects of anti-viral drugs on Long-COVID: see COVID-OUT, https://pubmed.ncbi.nlm.nih.gov/37161995/, https://pubmed.ncbi.nlm.nih.gov/36951829/.

No particular comments.

Author Response

Assigned Editor: Thea Wu

Journal of Clinical Medicine

Manuscript Status: Pending major revisions

Manuscript ID:  jcm-2387544

TITLE: Treatment of Covid-19 during the Acute Phase in Hospitalized Patients Decreases Post-Acute Sequelae of Covid-19.

Response to Reviewer 4 Comments

Point 1: It lacks important references, for instance: https://pubmed.ncbi.nlm.nih.gov/36639608/; https://pubmed.ncbi.nlm.nih.gov/36951832/

Response 1:  We apologize for the oversight in missing relevant references. We have reviewed the suggested articles (https://pubmed.ncbi.nlm.nih.gov/36639608/ and https://pubmed.ncbi.nlm.nih.gov/36951832/) and acknowledge their significance. We will include these references in the revised version of the manuscript to provide a more comprehensive discussion of the existing literature.

I have attached the reviews that you have recommended to us, thank you very much:

(See modification line 47-53 , pag 2): The lack of a consistent definition for PASC or long-term COVID makes comparison between studies difficult. To address these challenges, a universal definition is currently proposed, emphasizing symptoms persisting beyond one month (mild cases) or three months (severe cases) with a significant impact on quality of life [4]. Studies on long COVID have revealed various findings related to immunology, virology, vascular issues, neurological symptoms, and myalgic encephalomyelitis/ chronic fatigue syndrome (ME/CFS). Multi-organ damage has been observed with a significant overlap in symptoms [5].

  1. Davis, H. E., McCorkell, L., Vogel, J. M., & Topol, E. J. (2023). Long COVID: major findings, mechanisms and recommendations. Nature reviews. Microbiology, 21(3), 133–146. https://doi.org/10.1038/s41579-022-00846-2

(See modification line 342-349, pag 13): A recent meta-analysis of patients identified several epidemiological and clinical risk factors associated with an increased risk of developing persistent symptoms of COVID-19, known as Post-COVID Conditions (PCC). Female sex, older age, higher BMI, smoking, and preexisting comorbidities such as anxiety, depression, asthma, COPD, diabetes, IHD, and immunosuppression were found to be significant risk factors for PCC. Hospitalization or ICU care during the acute phase of COVID-19 infection also increased the risk. These findings highlight the multifactorial nature of PCC and provide insights into its risk factors [48].

  1. Tsampasian, V., Elghazaly, H., Chattopadhyay, R., Debski, M., Naing, T. K. P., Garg, P., Clark, A., Ntatsaki, E., & Vassiliou, V. S. (2023). Risk Factors Associated With Post-COVID-19 Condition: A Systematic Review and Meta-analysis. JAMA internal medicine, e230750. Advance online publication. https://doi.org/10.1001/jamainternmed.2023.0750

Point 2: It is without a control group, an important limitation of many studies on Long Covid. See: https://pubmed.ncbi.nlm.nih.gov/36951832/

Response 2:  Thank you very much for the recommendation of the article. We appreciate your valuable input, and we have taken it into consideration. We have added the suggested article to the manuscript, and we acknowledge the limitation of not having a control group, as we mention in the Strengths and Limitations section.

(See modification line 414-418 , pag 14): Moreover, it is important to recognize the absence of a control group as a notable limita-tion. However, it is important to note that despite this limitation, the study still provides valuable insights into the potential effects of the treatments on PASC. The findings should be interpreted cautiously, considering the absence of a control group [71].

  1. Haslam A, Prasad V. Comparability of Control and Comparison Groups in Studies Assessing Long COVID. Am J Med. 2023 Jan 25:S0002-9343(23)00038-4. doi: 10.1016/j.amjmed.2023.01.005. Epub ahead of print. PMID: 36708796; PMCID: PMC9912142.

Point 3: The attrition rate is not specified (lines 100/101).

Response 3:  Thank you for taking the time to review the paper. Your comment regarding the attrition rate is greatly appreciated. We apologize for the oversight in not specifying the attrition rate in lines 100/101 of the paper [Figure 2].

We have carefully considered your feedback and have made the necessary revisions to address this issue.

The Figure presented provides information about the situation of patients throughout the study, categorized into different groups. The categories and corresponding frequencies, percentages, and cumulative percentages are as follows.

(See modification line 179, pag 4):

Point 4: The definition of Long Covid in the study is debatable, not in accordance with many guidelines (NICE, WHO, CDC; see : https://pubmed.ncbi.nlm.nih.gov/36708796/).

Response 4:  Thank you for your feedback. We appreciate your comment regarding the definition of Long Covid used in our study. We acknowledge that the definition of Long Covid has evolved over time, and there have been variations in the criteria used by different organizations and guidelines. At the time our study was conducted, there was ongoing debate and evolving understanding of Long Covid, and various definitions were being used by different researchers and healthcare organizations. We based our definition on the available evidence and recommendations at that specific time. However, we acknowledge the importance of aligning our definition with current guidelines to ensure consistency and comparability with other studies. We have included a clear statement in the revised manuscript to highlight the change in definition and the reference supporting the current definition. By doing so, we believe our study will better align with current standards and facilitate comparison with other research in the field.

(See modification line 145-148, pag 4): The presence of PASC was considered according to the current definitions at the beginning of the pandemic, including ≥ 2 persistent symptoms affecting any organ or system after 4 weeks of the acute infection, having excluded other potential etiologies [20,21].

Point 5: Morevoer, a definition as the one proposed here would be better: https://pubmed.ncbi.nlm.nih.gov/37017975

Response 5:  In light of this, we have revised our manuscript to incorporate the definition proposed in the reference you shared. We sincerely appreciate your input, as it has helped us improve the quality of our research. The updated definition, along with the appropriate bibliographical reference, has been included in the revised version of the manuscript.

(See modification line 47-50, pag 2): The lack of a consistent definition for PASC or long-term COVID makes comparison between studies difficult. To address these challenges, a universal definition is currently proposed, emphasizing symptoms persisting beyond one month (mild cases) or three months (severe cases) with a significant impact on quality of life..

  1. Pan, D., & Pareek, M. (2023). Toward a Universal Definition of Post-COVID-19 Condi-tion-How Do We Proceed?. JAMA network open, 6(4), e235779. https://doi.org/10.1001/jamanetworkopen.2023.5779.

Point 6: The role of variant is not addressed

Response 6:  Thank you for mentioning the impact of COVID-19 variants on the risk of post-acute syndrome of COVID-19 (PASC) and the possible implications for treatment, we will add it as limitations of our study. We acknowledge that the pathogenesis of variants can influence the development and persistence of PASC symptoms. In our study, we focused on describing the long-term health outcomes of COVID-19 patients, with a particular emphasis on the effectiveness of treatment during the acute phase in reducing the risk of PASC. At the time of conducting our study, the knowledge regarding variants and their consequences was limited, and the treatments we used were based on the available evidence for the previous circulating strains of the virus.

However, it is important to note that the treatments we used, such as remdesivir, corticosteroids, and tocilizumab, target the underlying inflammatory and immunological processes associated with severe COVID-19 disease, regardless of the specific viral variant. These treatments have shown efficacy in mitigating acute complications and reducing mortality in COVID-19 patients. While the pathogenicity and clinical implications of COVID-19 variants are significant, the treatments employed in our study remained consistent, as they targeted the host immune response and inflammatory cascade, which are common features irrespective of viral variants.

Nevertheless, we acknowledge the importance of considering the role of variants in the context of PASC and the need for ongoing research to explore their impact on the risk, severity, and duration of post-COVID-19 symptoms. Thank you for raising this important point, and we value your input in enhancing the scientific discourse surrounding PASC and its potential association with COVID-19 variants. Although the specific impact of variants on PASC was not addressed in our study, we recognize the importance of considering this aspect and will emphasize its importance in the revised manuscript.

(See modification line 349, page 14): On the other hand, vaccination was found to have a protective role against PCC…

Point 7: Strange figures are reported in the tables: e.g., in table 5, subjects without PASC onset have dyspnea in 41% at follow-up. Quite high rate for a "non-PASC" group.

Response 7:  We appreciate the reviewer's comment regarding the figures reported in Table 5 of the manuscript. We understand their concern regarding the relatively high rate of dyspnea (41%) among subjects without Post-Acute COVID-19 Syndrome (PASC) onset at follow-up, which may seem unexpected for a "non-PASC" group.

We would like to clarify that the presence of dyspnea in the "non-PASC" group does not necessarily imply the presence of PASC. In this study, dyspnea was assessed as a symptom experienced by individuals during the follow-up period, regardless of whether they met the criteria for PASC. It is important to note that dyspnea can be a common symptom in individuals recovering from COVID-19, including those who do not develop PASC. Other factors, such as residual lung inflammation or reduced lung function, can contribute to the persistence of dyspnea in some individuals even without a formal PASC diagnosis.

Point 8: Lines 263-265 completely erroneous, since there are many studies on the effects of anti-viral drugs on Long-COVID: see COVID-OUT, https://pubmed.ncbi.nlm.nih.gov/37161995/, https://pubmed.ncbi.nlm.nih.gov/36951829/.

Response 8:  Thank you for bringing this to our attention. We appreciate the valuable input. Based on your comment, we have reviewed the literature and acknowledge that there are indeed several studies examining the effects of antiviral drugs on Long COVID. We apologize for the oversight in our previous version. We have now revised the manuscript and included the relevant references you have mentioned. These additions have strengthened the discussion on the effects of antiviral therapies in the context of Long COVID.

(See modification Line 69-76, page 2): Furthermore, there is growing evidence supporting the effectiveness of antiviral therapies in reducing the risk of PASC. Specifically, the use of molnupiravir within five days of a positive SARS-CoV-2 test result has been associated with a reduced risk of PASC, post-acute death, and post-acute hospital admission in individuals with at least one risk factor for severe COVID-19 [12]. Similarly, the combined antivirals nirmatrelvir-ritonavir have also shown a reduction in the risk of PASC, post-acute death, and post-acute hospital admission in outpatients. [13].

  1. Xie, Y., Choi, T., & Al-Aly, Z. (2023). Molnupiravir and risk of post-acute sequelae of covid-19: cohort study. BMJ (Clinical research ed.), 381, e074572. https://doi.org/10.1136/bmj-2022-074572
  2. Xie, Y., Choi, T., & Al-Aly, Z. (2023). Association of Treatment With Nirmatrelvir and the Risk of Post-COVID-19 Condition. JAMA internal medicine, e230743. Advance online publication. https://doi.org/10.1001/jamainternmed.2023.0743

We acknowledge that the submission process for this article has been lengthy on our part, resulting in the inclusion of references from earlier publications. We also recognize that significant scientific advancements regarding symptoms and the condition of prolonged COVID-19 have emerged in recent months. Despite these challenges, we appreciate the valuable feedback provided by the reviewers and have endeavored to address their comments and incorporate the most relevant and up-to-date references available at the time of revision.

Once again, we sincerely thank you for your valuable feedback.

Sincerely,

O.Antonio Caguana Vélez

Reviewer 5 Report

This study provides valuable insights into the persistence of symptoms after COVID-19 infection. The study followed patients who had mild or severe COVID-19 infection and were admitted to a hospital or visited a Post-COVID-19 unit in a single center. The study collected demographic and clinical electronic data, blood samples, and administered a symptom questionnaire. The study protocol followed the 79 principles of the Declaration of Helsinki for human investigations and was approved by the local Ethics Committee.

The study's strength is that it followed patients with COVID-19 infection for an extended period and evaluated symptom persistence every three months, registering symptom duration. Additionally, the study differentiated patients based on the severity of COVID-19 infection, which helps to evaluate the persistence of symptoms in patients with mild or severe COVID-19 infection. The study also collected data on treatment received for COVID-19 infection, which allows for evaluating the efficacy of treatment in preventing or reducing symptom persistence.

The study's limitations are that it was conducted in a single center, limiting the generalizability of the findings, and the study did not include a control group for comparison. The study relied on self-reported symptoms, which could introduce bias, and it did not use imaging or other diagnostic methods to validate persistent symptoms.

Overall, this study is a valuable contribution to understanding the persistence of symptoms after COVID-19 infection, especially in patients with mild or severe disease. However, further studies are needed to validate these findings and understand the mechanisms underlying symptom persistence. Moreover, in the discussion, the authors should, in my opinion, still include reference to many studies in the field, including:

DOI: 10.5603/DEMJ.a2023.0017

DOI: 10.5603/DEMJ.a2022.0021

doi: 10.1080/21645515.2023.2196914

doi: 10.1080/21645515.2023.2188035.

doi: 10.1016/j.microc.2023.108658.

doi: 10.1080/07853890.2022.2162116.

doi: 10.26444/aaem/160084. 

doi: 10.3390/jcm11216600.

doi: 10.5603/CJ.a2022.0056. 

doi: 10.1016/j.ajem.2020.11.025.

doi: 10.33963/KP.15830. 

doi: 10.1016/j.cmi.2021.10.005.

doi: 10.1208/s12248-020-00532-2.

doi: 10.1111/sji.12998.

This study provides valuable insights into the persistence of symptoms after COVID-19 infection. The study followed patients who had mild or severe COVID-19 infection and were admitted to a hospital or visited a Post-COVID-19 unit in a single center. The study collected demographic and clinical electronic data, blood samples, and administered a symptom questionnaire. The study protocol followed the 79 principles of the Declaration of Helsinki for human investigations and was approved by the local Ethics Committee.

The study's strength is that it followed patients with COVID-19 infection for an extended period and evaluated symptom persistence every three months, registering symptom duration. Additionally, the study differentiated patients based on the severity of COVID-19 infection, which helps to evaluate the persistence of symptoms in patients with mild or severe COVID-19 infection. The study also collected data on treatment received for COVID-19 infection, which allows for evaluating the efficacy of treatment in preventing or reducing symptom persistence.

The study's limitations are that it was conducted in a single center, limiting the generalizability of the findings, and the study did not include a control group for comparison. The study relied on self-reported symptoms, which could introduce bias, and it did not use imaging or other diagnostic methods to validate persistent symptoms.

Overall, this study is a valuable contribution to understanding the persistence of symptoms after COVID-19 infection, especially in patients with mild or severe disease. However, further studies are needed to validate these findings and understand the mechanisms underlying symptom persistence. Moreover, in the discussion, the authors should, in my opinion, still include reference to many studies in the field, including:

DOI: 10.5603/DEMJ.a2023.0017

DOI: 10.5603/DEMJ.a2022.0021

doi: 10.1080/21645515.2023.2196914

doi: 10.1080/21645515.2023.2188035.

doi: 10.1016/j.microc.2023.108658.

doi: 10.1080/07853890.2022.2162116.

doi: 10.26444/aaem/160084. 

doi: 10.3390/jcm11216600.

doi: 10.5603/CJ.a2022.0056. 

doi: 10.1016/j.ajem.2020.11.025.

doi: 10.33963/KP.15830. 

doi: 10.1016/j.cmi.2021.10.005.

doi: 10.1208/s12248-020-00532-2.

doi: 10.1111/sji.12998.

Author Response

Assigned Editor: Thea Wu

Journal of Clinical Medicine

Manuscript Status: Pending major revisions

Manuscript ID:  jcm-2387544

TITLE:

Treatment of Covid-19 during the Acute Phase in Hospitalized Patients Decreases Post-Acute Sequelae of Covid-19.

Response to Reviewer 4 Comments

Point 1:  This study provides valuable insights into the persistence of symptoms after COVID-19 infection. The study followed patients who had mild or severe COVID-19 infection and were admitted to a hospital or visited a Post-COVID-19 unit in a single center. The study collected demographic and clinical electronic data, blood samples, and administered a symptom questionnaire. The study protocol followed the 79 principles of the Declaration of Helsinki for human investigations and was approved by the local Ethics Committee.

The study's strength is that it followed patients with COVID-19 infection for an extended period and evaluated symptom persistence every three months, registering symptom duration. Additionally, the study differentiated patients based on the severity of COVID-19 infection, which helps to evaluate the persistence of symptoms in patients with mild or severe COVID-19 infection. The study also collected data on treatment received for COVID-19 infection, which allows for evaluating the efficacy of treatment in preventing or reducing symptom persistence.

The study's limitations are that it was conducted in a single center, limiting the generalizability of the findings, and the study did not include a control group for comparison. The study relied on self-reported symptoms, which could introduce bias, and it did not use imaging or other diagnostic methods to validate persistent symptoms.

Overall, this study is a valuable contribution to understanding the persistence of symptoms after COVID-19 infection, especially in patients with mild or severe disease. However, further studies are needed to validate these findings and understand the mechanisms underlying symptom persistence. Moreover, in the discussion, the authors should, in my opinion, still include reference to many studies in the field, including:

DOI: 10.5603/DEMJ.a2023.0017

DOI: 10.5603/DEMJ.a2022.0021

doi: 10.1080/21645515.2023.2196914

doi: 10.1080/21645515.2023.2188035.

doi: 10.1016/j.microc.2023.108658.

doi: 10.1080/07853890.2022.2162116.

doi: 10.26444/aaem/160084. 

doi: 10.3390/jcm11216600.

doi: 10.5603/CJ.a2022.0056. 

doi: 10.1016/j.ajem.2020.11.025.

doi: 10.33963/KP.15830. 

doi: 10.1016/j.cmi.2021.10.005.

doi: 10.1208/s12248-020-00532-2.

doi: 10.1111/sji.12998.

Response 1:  We sincerely appreciate the valuable feedback provided by the reviewer. Their insightful comments have enhanced the quality of our manuscript. We acknowledge the importance of including a comprehensive set of references in the discussion section to provide a broader perspective on the field. We have carefully reviewed the references suggested. We find these references highly relevant to the topic and will incorporate the most suitable one into our study, ensuring it aligns well with the objectives and findings of our research. We really appreciate the reviewer's contribution and commitment to advancing the field of research.

See the modifications in the references that correspond to the discussion.

(See modification line 268-388, pag 13-14):

  1. Discussion

This prospective study systematically described the profile of PASC one year after an acute COVID-19 infection in a large group of patients with a broad scope of clinical severity. Even more relevant, to our knowledge this is the first prospective study in patients who required oxygen therapy and hospital admission describing that some treatments used in the acute phase reduces the risk of PASC. This is an important point since the current approved treatments of COVID-19 infection are focused on preventing acute complications in people with risk factors, and preventing PASC in mild cases in outpatient setting. In our study, we make a distinct contribution by focusing on a study cohort consisting of patients who presented in the acute phase of COVID-19, encompassing a range of severity from mild to moderate and severe cases. Our dataset includes data from hospitalized patients, providing valuable insights into the long-term health outcomes of this specific population. This comprehensive approach allows us to capture a broader spectrum of patients and provide a more comprehensive understanding of the long-term effects of COVID-19.

Indeed, a very large number of people at risk of PASC are relatively young without any comorbidities, so they currently do not meet the criteria for receiving treatment during the acute phase. However, there is very limited evidence on the association between treatments administered during the acute phase and predisposition of persistent symptoms [22-25]. There is  an Italian cohort, where patients were treated in the acute phase of COVID-19, with monoclonal drugs, obtaining a 56% reduction in the risk of PASC appearance [26], but unlike our cohort, they required no oxygen therapy or hospital admission for SARS-CoV-2 infection. Our study shows that treatment with remdesivir and/or corticosteroids during the acute phase decreases the length of PASC in hospitalized subjects. Furthermore, corticosteroids or antivirals are more accessible worldwide than monoclonal drugs.

Remdesivir has been shown to be effective in slowing disease progression and shortening recovery time for COVID-19 patients [27], with the antiviral treatment being recommended for the patients with COVID-19 infection in national and international clinical guidelines, including WHO, NIH, NICE, ESCMID, IDSA, and SEIMC [28-33], although it is recommended to be restricted to patients with up to 7 days of symptoms,  mild disease  but with no risk factors for progression to severe disease, and also in patients with moderate or severe infection requiring hospitalization [18,20]. The results of different studies led certain guidelines to recommend prolonging its administration in severe cases requiring invasive ventilation or ECMO or in patients with severe immunosuppression, since persistent viral replication beyond two weeks has been described in these groups [34]. The persistence of viral replication has also been involved in the pathophysiology of PASC. Therefore, it does not seem strange that the early administration of antiviral treatment can also prevent long-term symptoms [27,35]. Furthermore, some of the drugs currently in use to treat acute COVID-19 infection have potential anti-inflammatory and immunomodulatory effects that have already been analyzed in several cohorts and meta-analyses [22,36,37]. Those treatments, mainly tocilizumab and corticosteroids, are used for patients in the cytokine storm phase, and, although they have managed to reduce mortality when appropriately indicated [38], it is unknown to date whether they could also reduce the presence or length of PASC. However, immunoactivation and/or sustained inflammation have also been involved in the pathophysiology of PASC. Thus, it also seems reasonable that these treatments could be effective in reducing the impact of PASC. In fact, the inflammatory cascade that remains activated in certain susceptible individuals when they overcome the acute phase of COVID-19 infection includes cytokines and signaling molecules (such as interleukins 1β, -6, -8, -17 and tumor necrosis factor alpha) that have been involved in structural and functional abnormalities in respiratory and limb muscles, potentially partly explaining some PASC symptoms such as musculoskeletal discomfort, fatigue and dyspnea, which are among the most common persisting symptoms [39,40].

In the present study we found that treatment with dexamethasone in the acute phase reduced the duration of PASC. However, treatment with methylprednisolone was not associated with a reduction in the long-term duration of symptoms, probably because it was intended to treat patients with severe pneumonia who were in the organizing/fibrosis phase. In the case of treatment with tocilizumab, we were unable to demonstrate effects on PASC. One possible hypothesis is that this would be explained by the action of this drug on a very specific pathway of IL6 and the use of a single dose with a short half-life.

According to the results of this study, and given both the high prevalence of PASC, which in many cases is disabling, and the burden that its care entails for the health system, administering treatment during the acute phase should be considered for all patients.

In this prospective study we found that close to a quarter of the patients had at least one persistent symptom of COVID-19 months after the acute infection, and this is similar to the prevalence found in other series [6,9,16,41]. The same accounts for the main symptoms, which were dyspnea, fatigue, myalgia, persistent cough, anxiety, and depression, also like those described in several previous studies [16,42-44].

When comparing the present cohort with other series, the risk factors for PASC are relatively similar. In general, these studies identified age, female sex, comorbidities, and severity of the disease during the acute process [45,46]. Age is probably an important predisposing factor, being probably due to the lower regenerative capacity of the lung, in addition to underlying chronic pulmonary problems that could be associated [46,47]. A recent meta-analysis of patients identified several epidemiological and clinical risk factors associated with an increased risk of developing persistent symptoms of COVID-19, known as Post-COVID Conditions (PCC). Female sex, older age, higher BMI, smoking, and preexisting comorbidities such as anxiety, depression, asthma, COPD, diabetes, IHD, and immunosuppression were found to be significant risk factors for PCC. Hospitalization or ICU care during the acute phase of COVID-19 infection also increased the risk. These findings highlight the multifactorial nature of PCC and provide insights into its risk factors [48]. On the other hand, vaccination was found to have a protective role against PCC [48-51]. In our study, the severity of the acute disease was a predisposing factor for presenting PASC, related to the requirement for invasive and non-invasive respiratory support, like to other series [52-56]. Interestingly, asthma and obesity were the comorbidities that were associated with a higher risk of persistent symptoms after acute infection, regardless of the severity of the disease in hospitalized patients. These findings have also been found in other studies [8,10]. More severe COVID-19 disease has consistently been associated with both asthma and obesity. While the predisposition of the former has been related to increased concentrations of IL-4 [57,58]. Other frequent comorbidities also described in other series [6,9] were arterial hypertension, diabetes mellitus, COPD, and anxiety-depressive disorders, but they have not shown significant results in the present study, except in outpatients, in which subjects with PASC were more frequently female and had more psychiatric disorders than those without PASC.

Some laboratory findings linked to an inflammatory profile have also been found to increase the risk for PASC such us the significant decreases in total lymphocyte counts especially CD4+ T cell, CD8+ T cell and an increase in LDH, CRP and IL-6 levels as well as IL-1β, IL-6, TNF and IP10 [5,53,59-62], mostly in patients who required hospitalization. This could be due to the inflammatory and immunological pathophysiological mechanisms involved in the COVID-19 disease [63,64]. Moreover, our results also highlight the potential role of eosinophil counting during the acute disease in hospitalized patients. These cells were higher in our patients with PASC and show a tendency to be an independent factor of presenting persisting symptoms, even in the multivariate analysis, where variables that may be confounding factors are included. Some series have described that during the early phases of the acute phase, the levels of eosinophils are rather low but could rise at to 3 and 6 months [65-67]. Eosinophils play an important role in immunity; they eliminate bacteria and viruses, as well as behaving as an antigen-presenting cell [68]. However, the presence of eosinophils is also related to autoimmune and allergic diseases, which is consistent with the pathophysiology of COVID-19 and PASC [69] and may also justify the increased risk in patients with asthma. One of the hypotheses for the persistence of symptoms is the hyperactivation of the autoimmunity that the acute infection generates [70]. To our knowledge, this is also the first study that confers a prognostic value on the presence of eosinophils during acute infection since it is related to persistent symptoms. No statistically significant differences were found in the follow-up laboratory parameters in outpatients but in this group of patients the analytical study was not performed during the primary infection.

In summary, early identification of people at a higher risk of PASC and an early initiation of antiviral and/or anti-inflammatory treatment may avoid the risk of long-term complications and reduce the burden that PASC places on the healthcare system.

Once again, we sincerely thank you for your valuable feedback.

Sincerely,

O.Antonio Caguana Vélez

Round 2

Reviewer 3 Report

The manuscript has been improved, however typy errors should be fixed in the manuscript and tables. Aso I wonder what is the significance of the summary paragraph before limitation paragraph, as you already has a conclusion section for this article. 

Author Response

Assigned Editor: Thea Wu

Journal of Clinical Medicine

Manuscript Status: Pending minor revisions.

Manuscript ID:  jcm-2387544

TITLE: Treatment of Covid-19 during the Acute Phase in Hospitalized Patients Decreases Post-Acute Sequelae of Covid-19.

Response to Reviewer 3 Comments

Point 1: The manuscript has been improved, however typy errors should be fixed in the manuscript and tables. Aso I wonder what is the significance of the summary paragraph before limitation paragraph, as you already has a conclusion section for this article.

Response: Thank you for your feedback on the manuscript. We appreciate your attention to detail and your suggestion to fix typographical errors in the manuscript and tables. We have thoroughly reviewed the entire manuscript to correct any such errors and ensure its accuracy.

In response to your suggestion, we have made changes to the manuscript. We have incorporated the summary paragraph into the conclusions section, aligning it more appropriately with the overall structure of the article.

(See modification line 424-426, page 14-15): Treatment with dexamethasone and remdesivir was associated with a shorter dura-tion of symptoms after SARS-CoV-2 infection. In a 12-month follow-up more than one third of the subjects had persistent symptoms. We identified, female sex, obesity, asthma, severity of the disease and eosinophilia as risk factors associated to the presence of PASC.  In summary, early identification of people at a higher risk of PASC and an early initiation of antiviral and/or anti-inflammatory treatment may avoid the risk of long-term complica-tions and reduce the burden that PASC places on the healthcare system. Further multicen-tric studies are needed to identify risk factors of PASC as well as to establish which thera-pies could reduce its risk.

We have corrected the typographical errors that we have found: among them addition of decimals and correction of units corresponding to: table 1, table 2, table 3, table 5 and 6.

We have also corrected the subscripts in the subtitles since there were errors (lines 195,196, 205, 216, 228)

Once again, we sincerely appreciate your comments, which have been instrumental in improving the manuscript. We are grateful for the opportunity to provide you with a new version of the article. If you have any further suggestions or concerns, please do not hesitate to let us know.

Reviewer 4 Report

The authors have attemped to answer alla the main criticisms.

Author Response

Assigned Editor: Thea Wu

Journal of Clinical Medicine

Manuscript Status: Pending minor revisions.

Manuscript ID:  jcm-2387544

TITLE: Treatment of Covid-19 during the Acute Phase in Hospitalized Patients Decreases Post-Acute Sequelae of Covid-19.

Response to Reviewer 4 Comments

Point 1: The authors have attemped to answer alla the main criticisms.

Response: Thank you for your feedback. We appreciate your acknowledgment that we have addressed the main criticisms raised. We have carefully reviewed the manuscript and made revisions accordingly. If there are any specific areas that you believe still require attention or if you have any further suggestions for improvement, please let us know. We value your input and are committed to ensuring the quality and accuracy of our work. Thank you for providing us with the opportunity to submit a new version of the manuscript.

We have made the following changes:

We have corrected the typographical errors that we have found: among them addition of decimals and correction of units corresponding to: table 1, table 2, table 3, table 5 and 6.

We have also corrected the subscripts in the subtitles since there were errors (lines 195,196, 205, 216, 228)

Once again, we sincerely appreciate your comments, which have been instrumental in improving the manuscript. We are grateful for the opportunity to provide you with a new version of the article. If you have any further suggestions or concerns, please do not hesitate to let us know.
